# A Promising Approach: Magnetic Nanosystems for Alzheimer’s Disease Theranostics

**DOI:** 10.3390/pharmaceutics15092316

**Published:** 2023-09-13

**Authors:** Catarina I. P. Chaparro, Beatriz T. Simões, João P. Borges, Miguel A. R. B. Castanho, Paula I. P. Soares, Vera Neves

**Affiliations:** 1Instituto de Medicina Molecular, Faculdade de Medicina, Universidade de Lisboa, 1649-028 Lisbon, Portugal; cchaparro@medicina.ulisboa.pt (C.I.P.C.); beatriz.simoes@medicina.ulisboa.pt (B.T.S.); macastanho@medicina.ulisboa.pt (M.A.R.B.C.); 2i3N/CENIMAT, Department of Materials Science, NOVA School of Science and Technology, NOVA University of Lisbon, Campus de Caparica, 2829-516 Caparica, Portugal; jpb@fct.unl.pt

**Keywords:** Alzheimer’s disease, drug delivery, magnetic nanoparticles, nanomedicine, theranostics

## Abstract

Among central nervous system (CNS) disorders, Alzheimer’s disease (AD) is the most prevalent neurodegenerative disorder and a major cause of dementia worldwide. The yet unclear etiology of AD and the high impenetrability of the blood–brain barrier (BBB) limit most therapeutic compounds from reaching the brain. Although many efforts have been made to effectively deliver drugs to the CNS, both invasive and noninvasive strategies employed often come with associated side effects. Nanotechnology-based approaches such as nanoparticles (NPs), which can act as multifunctional platforms in a single system, emerged as a potential solution for current AD theranostics. Among these, magnetic nanoparticles (MNPs) are an appealing strategy since they can act as contrast agents for magnetic resonance imaging (MRI) and as drug delivery systems. The nanocarrier functionalization with specific moieties, such as peptides, proteins, and antibodies, influences the particles’ interaction with brain endothelial cell constituents, facilitating transport across the BBB and possibly increasing brain penetration. In this review, we introduce MNP-based systems, combining surface modifications with the particles’ physical properties for molecular imaging, as a novel neuro-targeted strategy for AD theranostics. The main goal is to highlight the potential of multifunctional MNPs and their advances as a dual nanotechnological diagnosis and treatment platform for neurodegenerative disorders.

## 1. Alzheimer’s Disease

CNS diseases including neurological disorders account for significant public health concerns and in 2019 were the leading cause of disability-adjusted life years (349 million) and the second leading cause of deaths (nearly 10 million) [1].

Neurological disorders have enforced a heavy toll on the population due to life expectancy growth and population aging. These are crucial reasons to increase the demand for treatment and support services for rehabilitation in governments and healthcare systems [2,3].

CNS disorders include significant pathologies such as vascular damage, neuronal injury, neuroinflammation, and neurodegeneration. Within neurodegenerative disorders, AD is the second most prevalent, being the major cause of dementia (around 70% of dementia cases [1,4]). AD is a chronic and progressive brain disease that leads to the deterioration of cognitive function, most commonly impaired memory, and changes in thinking and behavior. It is predicted that at the current rate, 1 in 85 persons worldwide will be living with AD by 2050 [4].

The pathogenic process of AD can be linked to the damage and death of neurons that probably start decades before the disease’s clinical onset. During this mild cognitive impairment, neuritic dysfunction emerges in the brain’s hippocampus area, followed by the atrophy of the cerebral cortex and, lastly, dementia [5].

Despite the many theories for AD neuropathology, the most acceptable one is the amyloid beta (Aβ) protein-based hypothesis. This hypothesis comprises the buildup of cortical and cerebrovascular deposits of Aβ peptide, caused by Aβ protein aggregation [4,6]. Aβ peptide is an atypical proteolytic byproduct of the transmembrane amyloid precursor protein (APP) and is generated through α-, β-, and γ-secretases activity. The cleavage through α-secretase results in the release of a soluble amino (N)-terminal ectodomain (sAPPα) and C-terminal fragments (α-CTF). This process results in the “non-amyloidogenic” pathway since Aβ is blocked. The “amyloidogenic” pathway is triggered when APP is cleaved by a β-secretase at the N-terminal domain, which releases a soluble N-terminal fragment (sAPPβ) and the remaining C-terminal part (β-CTF). Both α-CTF and β-CTF are then cleaved in the transmembrane domain by a γ-secretase, yielding either extracellular 3 kDa peptide (p3) or Aβ, respectively [7,8]. Presenilin 1 or 2 (PS), nicastrin, anterior pharynx defective (APH1), and presenilin enhancer (PEN2) form the γ-secretase complex. The unspecific cleavage conducted by the γ-secretase enzymatic complex in the APP domain results in C-terminal truncated peptides ending with amino acids 37 to 43. These cleavages are roughly three amino acids apart: one at position 48 or 49, followed by another at amino acid 45 or 46, and culminating with a final cleavage at positions 38, 40, or 42. Aβ species ending with alanine at position 42 (Aβ42) have a higher potential to aggregate than Aβ40 and are hence considered the more toxic variant. Although almost 90% of the residues consist of Aβ40, the Aβ42 is the most abundant isoform for amyloid plaque development and leads to augmented neurotoxic effects [7,9,10].

The amyloid hypothesis predicts that genetic risk factors for AD include mutations in genes expressing APP and both PS proteins. Most of these mutations result in an overproduction or reduce clearance of Aβ42, leading to an abnormal accumulation of this form in the brain. The aggregation of Aβ42 monomers into oligomers, fibrils, and plaques triggers a cascade of neurobiological events such as inflammatory responses, aberrant buildup of hyperphosphorylated microtubule-associated tau (P-tau) protein, and other neuronal alterations. P-tau aggregates accumulate as intracellular neurofibrillary tangles (NFTs) and as dystrophic neurites associated with Aβ plaques [6,11,12]. Despite the observations of altered levels of Aβ, tau, and P-tau in cerebrospinal fluid (CSF), a growing body of evidence suggests that the development of senile plaques and neurofibrillary tangles in AD pathogenesis is yet unclear [13,14]. Senile plaques and neurofibrillary tangles are involved in neuroinflammation progress and the death of neurons, resulting in behavioral symptomatic alteration and memory loss. Recently, pathological and clinical data indicated immunological alterations associated with AD, including improved concentrations of pro-inflammatory cytokines in the blood and CSF [15]. Although inflammation within the brain, including improved reactivity of the microglia towards Aβ deposits, has been involved in the progression of the disease, the guarantee about these alterations as a reason for or a consequence of AD might be inconclusive [16].

Amyloid-β-derived diffusible ligands (ADDLs) are soluble spherical aggregates and appear as intermediates in the pathway of amyloid fibril formation. It was commonly thought that amyloid fibrils initiate a cascade of events that results in the known symptoms of AD, but recently attention has shifted towards the oligomeric soluble Aβ, which may be responsible for synaptic dysfunction [17,18]. However, further evidence suggests that the severity of cognitive impairments appears to correlate better with the loss of synapses rather than with the presence of Aβ deposits and NTFs in the brain [19,20].

Other theories involving vascular factors are emerging as elements that also underpin the disease. A decrease in cerebral blood flow (CBF), increased capillary tortuosity and IgG antibody trafficking, neurotoxic secretions from brain cells, and blood–brain barrier (BBB) disruption have all been detected to some extent in AD cases [21]. The earliest pathological signs of the disease seem to be more related to vascular factors than to Aβ accumulation, brain atrophy, and signs of cognitive decline [22,23].

The two phases of the vascular hypothesis suggest that AD is caused by a dual-stage progression. The first “hit” is related to pathologic conditions such as inflammation, oxidative stress, head injury, diabetes/metabolic syndrome, and vascular pathologies. This range of events triggers BBB dysfunction and oligemia (i.e., blood volume deficiency due to CBF reduction) via brain microcirculation damage. Besides the buildup of elevated levels of neurotoxic substances such as cytokines, BBB dysfunction becomes an obstacle for Aβ deposit clearance, while oligemia leads to an increase in Aβ production. In turn, the second “hit” consists of the accumulation of Aβ peptides and tau proteins in the brain. Together, these two stages are commonly attributed to the pathologies and symptoms associated with AD [24]. 

Different evidence-based studies point to cardiovascular diseases as the leading cause of AD dementia. Cerebral amyloid angiopathy (CAA), which is a predominant cause of BBB disruption and constitutes a pathological hallmark of AD, can trigger numerous vascular pathologies that lead to cognitive decline [25]. In preclinical AD, changes in vascular biomarkers occur before the development of cognitive impairment and before detectable increases in standard AD biomarkers, including amyloid deposition, decreased CSF levels of Aβ42, and increased CSF levels of tau and phosphorylated tau [26,27].

Based on evidence, BBB dysfunction may play a multifaceted role in AD both upstream and downstream of the amyloid cascade. It is plausible that the deficient BBB efflux of Aβ peptides could trigger the amyloid cascade and simultaneously be caused by this sort of event. Pathologic states that are also considered risk factors for AD have been identified to alter the function of many BBB transporters [28]. Defects of Aβ transporters caused by the loss of BBB integrity can likely result in Aβ peptide accumulation in the CNS. Aβ has a much higher propensity to transition to β-sheet conformation and aggregate as its concentration increases [29]. Optimal BBB efflux occurs for Aβ monomers; however, with increased β-sheet content and further aggregation, the affinity of transporters decreases, and Aβ accumulation in the CNS hinders its efflux [30,31]. Alterations/breakdown of the transport barrier are caused by the transporters’ expression alterations, and it has been proved that some transporters are downregulated in AD [32]. Aβ accumulation can modulate transporter functions during the disease and promote a microvascular decrease in its expression. 

It is clear that the role of Aβ in the CNS can be a cause or a consequence of BBB dysfunction in AD; however, there are other pathologies independent of Aβ activity that can phenotypically mimic BBB disruption. These suggestions highlight the complexity of AD, and the possibility that AD has diverging etiologies that converge in Aβ and tau accumulation. Despite the uncertainty that lies in the evidence of BBB disruption in AD, it is considered that alterations in transport and communication within the cells that regulate the BBB are the most prominent affected functions in the barrier and may become impaired in AD [21].

## 2. Alzheimer’s Disease Diagnosis

Despite the lack of a disease-altering treatment, earlier diagnosis of AD would provide more time for patients to plan for support services in the later stages of the disease and also to participate in early intervention clinical trials [33,34]. In addition to preventing irreparable neuronal damage, an early and precise diagnosis would avoid the administration of ineffective approaches for AD and spare patients from extensive testing [35].

The conventional diagnosing methods of AD focus on the examination of Aβ deposition, pathologic tau, and neurodegeneration through analysis of biomarkers in the CSF [36], positron emission tomography (PET) [37], and magnetic resonance imaging (MRI) [38]. Liquid biopsy techniques and blood base biomarkers are reviewed elsewhere [39,40]. PET is the most frequently used diagnostic technique for Aβ plaque detection, with high specificity and sensitivity using radiotracers. Amyloid PET is currently used in clinics with three Aβ tracers approved by the US Food and Drug Administration (FDA): Florbetapir F-18 (Amyvid) [41], Florbetaben (Neuraceq, Piramal Imaging) [42,43], and [^18^F]-flutemetamol (Vizamyl R^®^) [44]. One-third of AD diagnoses result from amyloid PET imaging. Moreover, tau-binding PET tracers recently developed exhibit a high binding affinity with tau deposits and are strongly related to cognitive impairment, atrophy, and amyloid PET markers. Nevertheless, tau PET is still in its early phases of development and is hampered by off-target binding, found both in vivo and in vitro for all tau tracers [45].

High cost, low availability, and radiation exposure are the drawbacks associated with diagnosing AD through PET. Additionally, the implementation of strict criteria for its proper use is essential due to the observation of positive results in healthy controls [46]. Individual plaque imaging may be limited by low spatial resolution, making it difficult to detect the initial stages of amyloid formation. Moreover, according to the Amyloid Imaging Task Force of the Alzheimer’s Association and Society for Nuclear Medicine and Molecular Imaging, amyloid PET is only recommended for use in individuals where there is substantial doubt about the pathology underlying proven cognitive impairment [46]. MRI, on the other hand, is a cost-effective and widely available alternative to PET.

Because of its greater soft tissue contrast, MRI is the ideal technique for non-invasively monitoring of the development of a disease or the impact of its treatment. MRI brain scans use magnetization transfer between the protons from brain tissue macromolecules and the protons in water molecules. Typically, brain MRIs comprise multi-planar images of the entire volume of the brain, magnetic resonance angiography, as well as diffusion and chemical shift imaging. These techniques usually take advantage of the brain’s natural contrast, which emerges from differences in T1 and T2 relaxation times of different types of tissue, particularly with higher values for CSF. T1, or longitudinal relaxation time, ranges from a few milliseconds to several seconds in biological tissues, while T2, or transverse relaxation, is usually shorter than T1 and can occur independently or in conjunction with T1 relaxation. External factors, such as magnetic inhomogeneity or susceptibility artifacts in tissues, can decrease the real value of T2 relaxation time, which is known as T2* [47]. MRI is routinely used to identify vascular lesions and atrophy in AD; however, it has the potential to image Aβ plaques both in humans and animals. Despite that, the use of endogenous contrast has, until recently, only been successful in older animals with amyloid plaques larger than 50 nm in diameter and greater iron loads. Long imaging times and ultra-high-field scanners are required for these procedures, which are not compatible with clinical imaging [35,48,49,50]. Instead, contrast-enhanced MRI proves to be efficient in a specific and sensitive AD diagnosis. The use of targeted exogenous contrast agents, which can reduce both T1 and T2 relaxation times, provides evidence as a potential tool for the early detection of amyloidosis.

MRI contrast agents that act on T1 and T2, respectively, are separated into two distinct categories. Paramagnetic nanoparticles (NPs) containing lanthanide elements such as gadolinium (Gd^3+^) affect the T1 relaxation times and have been traditionally used as “positive” contrast agents, preferentially fastening the T1 recovery with contrast-enhanced regions appearing bright or “hyperintense” in T1-weighted images [51]. T2 contrast agents, on the contrary, can shorten the T2 relaxation period, which reduces both T2 and T2* signals and causes a dark contrast or “hypointense” in T2-weighted images. Ferromagnetic and superparamagnetic iron oxide nanoparticles (SPIONs) are negative contrast agents and take effect in T2 relaxation time, decreasing the MRI signal in the regions of their administration [7,52].

Gadolinium-based contrast agents are usually presented as biocompatible chelates of Gd^3+^ and demonstrate the Aβ plaque imaging both ex vivo and in vivo. Gd-based contrast agents are commercially available; however, their use is related to toxic side effects since ionic Gd complexes can release Gd^3+^ ions, leading to an increased risk of nephrogenic system fibrosis development after its accumulation in tissues [51]. As an alternative, SPIONs are seen as MRI contrast agents that are much safer than Gd.

### SPIONs as Contrast Agents for AD Diagnosis

Currently, pure iron oxides such as magnetite (Fe_3_O_4_) and maghemite (γ-Fe_2_O_3_) are the most common biocompatible and biodegradable magnetic nanomaterials which fulfill a large number of biomedical needs. Iron oxides are nontoxic, tolerated biologically, and can be integrated into human natural processes of iron metabolism after injection, serving as MRI contrast agents or drug delivery systems. Besides these benefits, SPIONs offer other important biomedical applications such as iron supplementation, in vivo cellular and molecular labeling, protein separation, and magnetic hyperthermia [53,54,55,56].

SPIONs are well known for their ability to generate heat when subjected to an external alternating magnetic field (AMF). The combination of magnetic hyperthermia and chemotherapy brought SPIONs into view as potential nanocarriers for cancer treatment. Regarding brain cancers, glioblastoma multiforme (GBM) is the most predominant and lethal primary intrinsic brain tumor [57]. SPIONs have been broadly investigated for their potential to exert antitumor effects in GBM through magnetic hyperthermia and guidance [58,59,60,61]. For instance, Carvalho et al. [58] studied the effect of AMF application on glioma cells after treatment with SPIONs-based polymersomes conjugated with doxorubicin (DOX). The exposure of these nanocarriers to an AMF led to a significantly higher cell death response due to the synergistic effect of DOX release and the heat generated in glioma cells. Additionally, Cui and colleagues [61] developed a dual-targeting strategy combining magnetic guidance and transferrin receptor-binding peptide T7 to overcome the BBB and actively co-deliver curcumin (CUR) and paclitaxel (PTX) in the brain. The T7-modified magnetic-polymeric NPs were prepared by co-encapsulation of SPIONs and both CUR and PTX. SPION encapsulation offers the possibility to magnetically guide the system and, subsequently, to improve its brain accumulation in an orthotopic glioma-bearing mouse model. In vivo MRI revealed a five-fold increase in brain delivery over nontargeting NPs. Furthermore, the application of a magnetic field increased the anti-glioma therapy efficacy, with all animals bearing orthotopic glioma surviving, compared to a 62.5% survival rate for the combined group receiving free CUR and PTX.

In addition to magnetic hyperthermia, SPIONs can improve MRI sensitivity when used as in vivo or in vitro contrast agents, revealing great potential as diagnosis and therapeutic monitorization platforms [62]. While the nephrotoxicity of Gd NPs is a cause of concern, it is hypothesized that SPIONs are included in the biological iron reserves after a few days in the system [63]. For that reason, they can be applied as detection tools for the follow-up of therapeutic interventions. SPIONs with FDA approval have core diameters ranging from 50 to 200 nm, good biodistribution, and biocompatibility. Some of them were approved for clinical usage as MRI contrast agents, including Feridex^®^, Clariscan^®^, and Resovist^®^, before being withdrawn from markets due to low commercial interest [47,64].

A breakthrough towards AD prevention can be made by combining their diagnostic properties with endogenous therapeutic molecules. Particularly, targeted SPIONs have been widely used to detect Aβ plaques of AD both in vitro and in vivo. In a study performed by Wadghiri et al. [65], Aβ_1–40_ peptides were absorbed onto dextran-coated monocrystalline iron oxide NPs (MIONs), and the functionalized particles enabled the detection of numerous plaques by MRI, with an affinity for Aβ_1–42_ peptides (*K*_D_ = 202 nM) comparable to that of free Aβ_1–40_ (*K*_D_ = 266 nM). Later, the same authors reported the functionalization of poly(ethylene) glycol (PEG)-coated ultrasmall SPIONs (USPIONs) with Aβ_1–42_ peptides [66]. USPIO-PEG-Aβ_1–42_ nanoparticles were injected intravenously in AD transgenic mice without mannitol co-injection. The particles allowed a good contrast to image amyloid deposits by µMRI without the need for an agent to improve BBB permeability.

Derivatives of Aβ peptides have been used as targeting moieties. Heptapeptides PHI (C-FRHMTEQ-C) and PHO (C-IPLPFYN-C) exhibit a high affinity for Aβ_1–42_ peptides [67]. SPIONs conjugated with PHO peptides were able to cross the BBB and accumulate in the brain 90 min after injection, improving the contrast in MRI. SPIONs-PHO are characterized by a slow blood clearance; however, they are eliminated faster than P1-grafted SPIONs and SPIONs-PEG, which have a higher blood retention time. One week after injection, none of the SPION derivatives were found in mice organs, and no in vivo toxicity was observed [68].

Curcumin, for instance, is a natural compound that can bind to Aβ plaques specifically. Chang et al. [69] injected CUR-conjugated SPIONs (CUR-MNPs) into Tg2576 mice and non-transgenic mice and observed that the CUR-MNPs can penetrate the BBB of the transgenic AD model and effectively bind amyloid plaques. Ex vivo T2*-weighted MRI revealed more dark spots in AD mice brains, but no plaques were found in the control groups, which are aligned with the Aβ plaques on immunohistochemically stained sections.

Another study suggested that both oligomer-specific single-chain variable fragment (scFv) antibodies (W20) and a class A scavenger receptor (SR-A) activator (XD4) are promising diagnostic probes for early-stage AD when used to functionalize SPION surfaces (W20/XD4-SPIONs). These multifunctional magnetic NPs could recognize Aβ oligomers (AβOs) and promote microglial AβOs phagocytosis due to the property of XD4 for SR-A activation. W20/XD4-SPION accumulation in the brain provided the distinction between AD transgenic mice from wild-type (WT) controls due to the enhanced MRI contrast of pathological AβOs regions in brains [70].

Furthermore, SPIONs proved to be suitable as MRI detection tools after therapeutic interventions. Hour et al. [71] determined the effects of the delivery of human Wharton’s jelly-derived mesenchymal stem cells (MSCs) (WJ-MSCs) in the cognitive improvement of rats after injection towards the hippocampal area of AD models. For that purpose, WJ-MSCs were labeled with dextran-coated SPIONs (Figure 1A,C) and their presence in the hippocampal area was confirmed by MRI, where the signal intensity was reduced by increasing the number of cells (Figure 1B,D). Also, behavioral studies and histological assessments revealed cognitive and hippocampal cell functionality improvement, respectively.

In addition to Aβ detection and therapeutic response monitoring through longitudinal analyses, SPIONs are highlighted as promising drug delivery systems. Mahmoudi et al. [72] studied the physicochemical properties of SPIONs on the Aβ fibrillation kinetics and demonstrated that both the surface charge and thickness of the coating layer of SPIONs (Figure 2A) promote a “dual” effect on the fibrillation process in aqueous solutions. Lower SPION concentrations decreased the Aβ fibrillation rate, while higher concentrations increased it. Concerning the coating charge, it was observed that positively charged SPIONs promoted fibrillation at significantly lower NP concentrations compared with negatively charged or uncharged SPIONs. Both negative and plain NPs (single and double layer) decreased fibril size and promoted a narrower size distribution. On the contrary, positively charged SPIONs led to the formation of fibrils with a very broad size distribution (Figure 2B).

Moreover, Mirsadeghi et al. [73] explained the effect of SPIONs (magnetized or non-magnetized with different coating molecules) on Aβ fibrillation kinetics (Figure 2C–E) and observed that under a magnetic field, both positively (SPIONs-PEG-NH_2_) and negatively charged SPIONs (SPIONs-PEG-COOH) promote fibril formation in aqueous media at low concentrations. However, the positively charged/magnetized SPIONs accelerate the fibrillation process compared with uncharged NPs or SPIONs-PEG-COOH. Additionally, for high concentrations of PEG-NH_2_-coated SPIONs, the fibrillation process is accelerated even without the magnetic field application, while at lower concentrations this effect is inhibited.

These findings lead to the hypothesis that SPIONs designed for medical imaging applications should consider being coated with negatively charged or uncharged molecules, as these charge-like properties diminish undesired side effects such as protein fibrillation and guarantee the proper magnetic functions of SPIONs [7]. For instance, Amiri et al. [74] demonstrated how the formation of protein corona (PC) on SPION surfaces led to a slight increase in the relaxivity of negatively charged SPIONs, while drastically decreasing that effect in the positively charged ones, which were entirely related to particle agglomeration in the presence of proteins.

## 3. Protecting SPIONs: Application of Different Coating Molecules

The coating of SPIONs primarily aims at protecting the magnetic core from oxidation, so that magnetism can be preserved for a long time. Additionally, the nanoparticulate is protected from aggregation, biodegradation, structural alterations, or co-assembly with components of biological systems [75]. To grant SPIONs the ability to target and release drugs specifically, it is crucial to change their surface charge and hydrophobic character, and for that several types of surface coatings have been developed for SPIONs as drug delivery systems (DDSs).

### 3.1. Small Molecules

SPION surfaces are composed of numerous hydroxyl (OH^−^) groups, allowing small molecules and surfactants to be attached. The presence of these molecules at the surface of nanoparticles enables them to retain their original magnetic properties, maintain a small hydrodynamic diameter, and improve their hydrophilicity simultaneously [76,77]. Ligand- and phase-exchange reactions are the most conventional methods for small molecule modifications. Catechol, sulfate, carboxylic, phosphate, and citrate have a strong affinity to SPIONs, and therefore they can be exchanged with the pre-attached surface of organic groups like oleic acid or oleylamine. These molecules can also be used as tail-end groups of polymers such as PEG or polyethyleneimine (PEI) [78].

Silane is one of the most widely studied small coating molecules and can be covalently bound to the hydroxyl groups at SPION surfaces using the alkoxysilane reaction (–Si–O–R, where R is commonly –CH_3_ or –CH_2_–CH_3_). Further crosslinking events produce a thin inorganic silica layer around the particles, turning the initially hydrophobic SPIONs into hydrophilic ones, reducing aggregation, and improving stability [79]. Modified silanes, such as 3-aminopropyltriethyloxysilane (APTES), p-aminophenyl trimethoxysilane mercaptopropyltriethoxysilane (MPTES), and 2-(carboxymethylthio) ethyltrimethylsilane, are often used for transferring –NH_2_, –SH, and –COOH groups to naked iron oxide NPs, respectively, which is convenient for further modification with drugs or targets [78,80].

Jordan et al. [81] demonstrated the effect of dextran and aminosilane-coated SPIONs as thermotherapeutic agents on rat malignant glioma. The result of thermotherapy revealed that aminosilane-coated SPIONs led up to a 4.5-fold prolongation of survival over controls, while dextran-coated particles did not indicate any advantage. Moreover, Sillerud et al. [82] developed an MRI nanosystem for AD diagnosis based on the functionalization of anti-Aβ protein precursor (AβPP) with aminosilane-coated SPIONs. The authors confirmed that the anti-AβPP conjugated SPIONs were able to cross the BBB and act as a contrast agent for MRI of amyloid plaques. The conspicuity of the plaques increased from an average Z-score of 5.1 ± 0.5 to 8.3 ± 0.2 when the plaque contrast-to-noise ratio was compared in control AD mice with AD mice treated with magnetic NPs, indicating that the nanosystem crossed the BBB. Additionally, the number of MRI-visible plaques per brain increased from 347 ± 45 in the control AD mice to 668 ± 86 in the magnetic NP-treated mice.

### 3.2. Polymeric Coating

In recent years, polymer-coated SPIONs have drawn much more attention owing to their widespread applications in various research areas, including nanomedicine. To date, many polymer coatings have been developed via surface coverage or through micelles formation. Dextran, chitosan, PEG, polyvinyl alcohol (PVA), polydopamine (PDA), polyethyleneimine (PEI), polyvinylpyrrolidone (PVP), polyamidoamine (PAMAM), and the copolymer poly (lactic-co-glycolic acid) (PLGA) are common surface coverage polymers used in SPION coatings [83]. To combine the advantages of different polymers, copolymers were also developed. For instance, copolymers of PEG and PEI have both the ability to load genes and prolong particle half-life in blood [84], whereas copolymers of PLGA and PEG can help nanoparticles escape from the endo-lysosomal compartment to the cytoplasmic compartment and reduce the hydrophobicity of PLGA [85]. Moreover, stimuli-sensitive amphiphilic block copolymers have been designed to control the release of drugs [86].

#### 3.2.1. Dextran

Dextran is a polysaccharide with excellent biocompatibility that is soluble in water, and its coating onto SPION surfaces has a significant impact on their physicochemical properties. Dextran-coated SPIONs were first described in 1982 by in situ technology and approved by the FDA in 1996 (Feridex^®^) [87]. Dextran and its modifications were used in the study of brain diseases. A combination of dextran sulfate-coated SPIONs and quercetin was shown to be less toxic to PC12, a cell line that rapidly and reversibly responds to nerve growth factor (NGF) [88], than dimercaptosuccinic acid (DMSA)-coated SPIONs. In detail, dextran sulfate-coated SPIONs in a concentration lesser than 50 µg/mL had no significant toxicity to PC12 cells, while 1.5 mM of DMSA-modified SPIONs was shown to be toxic to those cells [89]. In Kouyoumdjian et al. [90] study, dextran coats the external surface of SPIONs, forming a stable colloidal suspension, and after modification with a sialic acid methyl ester derivative, the glyconanoparticles (NP-Sia) were purposed for the detection of Aβ plaques through MRI and Prussian blue staining (Figure 3A). The superparamagnetic nature of NP-Sia enabled the ex vivo detection of amyloid plaques by MRI, where dark spots were observed on the surface of Aβ brains incubated with the glyconanoparticles (Figure 3B(a)). Prussian blue staining was used to support the MRI results, where the presence of a blue color indicates the areas bearing iron oxide NP. This staining showed that only Aβ brains treated with NP-Sia exhibited the characteristic blue color (Figure 3B(e)), corroborating the MRI detection. Both dark spots and the blue color disappeared along the following conditions since free sialic acid was added during incubation to compete with NP-Sia binding (Figure 3B(b,f), respectively). Also, Aβ brains without NP incubation or normal mouse brains incubated with NP-Sia were used as controls to show that NP-Sia were able to bind specifically to Aβ (Figure 3B(c,d,g,h)). Moreover, NP-Sia converted Aβ to its less toxic form and ensured the protection of SH-SY5Y neuroblastoma cells from Aβ-induced cytotoxicity (Figure 3C).

#### 3.2.2. Chitosan

Chitosan (CS) is a biodegradable and biocompatible polymer that is considered a promising polymeric NP-based carrier for brain drug delivery. CS NPs have some unique features, such as a mucoadhesive nature and intrinsic bioactivity, which can not only promote the penetration of drugs into the brain through the olfactory route but also represent anti-AD therapeutics themselves [91,92]. Hassanzadeh et al. [93] studied the bioactivity and neuroprotective effect of both magnetic and non-magnetic CS NPs loaded with tacrine in AD rats. Tacrine was the first cholinesterase inhibitor approved by the FDA to treat the symptoms of mild to moderate AD. The tacrine-loaded CS NPs revealed the ability to prevent a behavioral decline in AD by improving special learning and memory. In addition, it was also demonstrated that the incorporation of magnetic NPs increased seladin-1 levels compared to the non-magnetic particles. Concerning that, it was possible to associate this increase with neurodegeneration resistance, since seladin-1 is a neuroprotective gene identified and found to be down-regulated in AD-vulnerable brain regions [94]. Based on these results, the authors suggested that the magnetic approach is highly promising for future studies, as it not only enhances the BBB penetration of tacrine but is also able to selectively deposit the drug into target brain regions, contributing to even higher bioactivity at the action site than non-magnetic chitosan NPs.

Regarding brain tumors, a magnetic multifunctional nanosystem was developed to treat glioblastoma. CS-SPIONs containing antitumor DOX and fluorescent dye Rhodamine B showed improved cell uptake and cell killing by inducing a concurrence of cell apoptosis and autophagy in the treated tumor U251 cells when conjugated with a tumor-specific ligand-transferrin (Tf). Moreover, the results showed that the fabricated Tf-functionalized CS-SPION nanocarriers demonstrated immediate responses under magnetic fields [95].

#### 3.2.3. Poly (Ethylene Glycol) PEG

PEG is another frequently used water soluble polymer like dextran. Several methods and approaches have been reported to synthesize PEG-coated SPIONs over time, mainly for biomedical applications [96,97]. PEG is synthesized by the anionic ring-opening polymerization of ethylene oxide, and it is considered a “shield” molecule against NP aggregation and opsonization and reduces their uptake by macrophages. PEGylation also extends NPs’ blood circulation time in vivo, making them suitable for targeted therapy [98]. PEGylated-SPIONs were used to study the cellular delivery of small interfering RNA (siRNA) against the expression of the β-site APP-cleaving enzyme 1 (BACE1) gene, which is thought to be related to the accumulation of AβAPP products as a consequence of its up-regulation. The co-immobilization of siRNA and the translocating outer membrane protein A (OmpA) on PEGylated-SPIONs enhanced the nanoparticulate cellular uptake through endocytosis, where endosome formation in SH-SY5Y cells was probably escaped due to the proton-sponge effect characteristic of PEGylated NPs or by a translocation mechanism in the case of OmpA function (Figure 4A–E). Moreover, the biological activity of the siRNA-immobilized molecules was maintained, as evidenced by the successful silencing of the BACE1 gene in HFF-1 cells (Figure 4F) [99].

Li et al. [100] successfully formulated antibiofouling polymer PEG-block-allyl glycidyl ether (PEG-*b*-AGE)-SPIONs for the detection of Aβ peptides and tau proteins in AD through liquid biopsy (Figure 5A). The prepared PEGylated-SPIONs efficiently suppressed non-specific interactions with Aβ peptides and tau proteins, and when conjugated with the capturing antibody, showed improved specificity (>90%) and sensitivity (>95%) (Figure 5B–E). Also, these novel antibody-conjugated antibiofouling magnetic nanoparticles demonstrated a better performance in capturing Aβ peptides and tau proteins from blood samples over common magnetic separating agents (Dynabeads^®^) (Figure 5F,G).

#### 3.2.4. Poly(Lactic-co-Glycolic Acid) PLGA

PLGA is a copolymer of poly(lactic acid) (PLA) and poly(glycolic acid) (PGA). It is highly biocompatible and it is already approved by the FDA and the European Medicines Agency (EMA) as a drug delivery vehicle for parenteral administration, diagnosis, and other applications of basic and clinical research, including cancer, cardiovascular diseases, tissue engineering, and vaccines [101]. Various types of block copolymers of PLGA with PEG have been developed, such as PLGA-PEG or PEG-PLGA-PEG. SPION-loaded PEGylated-PLGA NPs were used for PTX delivery as an anti-glioma drug. BBB disruption in the GBM area was observed through MRI and an ex vivo biodistribution study showed enhanced accumulation of NPs in the brain of GBM-bearing mice with magnetic targeting. PTX-PEG/PLGA-SPIONs showed less toxicity than free PTX in vivo and the magnetic targeting prolonged the median survival time when compared to passive targeting and control treatments [102].

In another approach, Rodriguéz et al. [103] proposed zinc-doped magnetite Zn_x_Fe_3−x_O_4_ NPs (ZnFeNPs) encapsulated in a PLGA matrix, generating polymeric magnetic beads (MBs) that afterward were covered with PEI (MB@PEI). These MBs were developed as electrochemical immunosensing platforms for AD diagnosis. The conjugation of affinity protein neutravidin (NAV) to MB@PEI promoted a higher saturation magnetization compared to commercially available NAV-modified MBs and also a better reproducibility (giving a relative standard deviation of 4% for MB@NAV and 12% for commercial MBs). MB@NAV was then applied to tau protein detection with a detection limit (LOD) of 63 ng/mL and demonstrated an excellent performance in human serum samples.

### 3.3. Lipid Coating

Lipids, as the constituents of cellular membranes, provide a biocompatible protective barrier for NPs. Lipids can form a closed double-layer structure in an aqueous solution containing hydrophilic and hydrophobic space and are one of the most popular DDSs [78]. For instance, the mRNA vaccine is a star in controlling SARS-CoV-2 infection, where lipid NPs not only protect mRNA from hydrolysis by nuclease, but also help mRNA penetrate through biological barriers (i.e., blood circulation, the cell membrane bilayer, and the endosomal trap) to successfully express target proteins intracellularly [104,105].

Magnetic liposomes emerged as a theranostics platform and present interesting properties regarding their high resistance against intracellular degradation compared with the known used citrate- or dextran-coated SPIONs [106]. Numerous studies have been conducted concerning the brain application of distearyl phosphatidylethanolamine (DSPE)-PEG-based iron liposomes. DSPE-PEG-formed SPIONs were developed by Hu et al. [107] to specifically detect amyloid plaques by MRI, achieve the drug-controlled release of AD therapeutic agents through H_2_O_2_ response, and prevent oxidative stress. In this study, the proposed NPs were co-loaded with Congo red and Rutin molecules. Congo red has been used as a histochemical stain for stacked β-sheet structures, and so in Aβ deposit quantification. On the other hand, Rutin is a glycone of quercetin with a flavonol structure with a powerful antioxidant effect. The authors suggested that combining the co-loading of Congo red and Rutin molecules with the magnetic liposome structure could promote a great MRI contrast upon specifically binding to amyloid plaques, while reducing both Aβ aggregation and neurotoxicity. The results showed that Congo red/Rutin-DSPE-PEG-formed SPIONs (Congo red/Rutin-MNPs) could inhibit the Aβ-induced cytotoxicity and increase SH-SY5Y cell survival from 67% to 92% as a result of the neuroprotective effects of Rutin. In vivo results showed that the particles’ administration resulted in a more specific binding/detection of amyloid plaques and gave a greater contrast-to-noise ratio on an MRI (Figure 6A(I,II)). Moreover, Congo red/Rutin-MNPs could significantly rescue memory deficits and ameliorated neurologic changes in AD transgenic mice (Figure 6B).

Moreover, Ruan et al. [108] developed a nanotheranostics system consisting of CUR and SPIONs encapsulated in DSPE-PEG modified with CRT (CRTIGPSVC) and QSH (QSHYRHISPAQV) peptides, which promotes specific binding to TfR and early Aβ plaques in the brain, respectively (SDP@Cur-CRT/QSH). The SDP@Cur-CRT/QSH nanosystem enables efficient CUR delivery to the brain for sensitive AD diagnosis and amyloid plaque clearance. It demonstrates peptide-targeted BBB penetration, precise delivery to Aβ plaques for sensitive therapeutic monitoring via MRI, and cognitive improvement attributed to neuroprotection and neurogenesis induced by BDNF. Additionally, the proposed nanosystem inhibits the Aβ plaque burden through the inhibition of the NLR family pyrin domain containing 3 (NLRP3) inflammasomes.

## 4. Nanotechnology Approaches for Brain Drug Delivery: Focus on Alzheimer’s Disease Theranostic

### 4.1. Strategies to Overcome the BBB

The limited and regulated passage of molecules from the periphery into the brain parenchyma constitutes a major reason why the current therapeutic molecules for AD fail to act effectively on the damaged areas of the brain. Almost 98% of small molecule drugs and virtually all the large ones are routinely excluded from the brain due to the innate resistance of the BBB [109]. The hindering of therapeutic agents’ permeation incites the use of receptors and transporters expressed at the luminal membrane of the BBB endothelium as potential vehicles for these molecules’ transportation into the brain [110].

Receptor-mediated transport (RMT) and adsorptive-mediated transport (AMT) belong to the ATP-binding cassette (ABC) transporters class and are responsible for the efflux of macromolecules from the CNS using endocytic vesicles to promote their cross through the BBB [111,112].

The concept of AMT through the BBB began with the observation that brain uptake of polycationic proteins did not involve binding to the endothelial cell surface [113,114]. Histone, avidine, and cationized albumin are some examples of macromolecules that interact with the negatively charged cell surface and consequently trigger transcytosis and exocytosis towards the abluminal surface of the brain endothelial cells [115]. Therapeutically, AMT can be achieved in one of two ways: (i) by building cationic surface charge into the drug or NPs, or (ii) by conjugating the drug or NPs (usually covalently) with a positively charged moiety, such as a cell-penetrating peptide (CPP) [116]. CPPs are short targeting vectors that typically consist of fewer than 30 amino acids, and the only common feature of these peptides appears to be that they are amphipathic and net positively charged, thus allowing them to penetrate plasma membranes and transport their cargo into cells [117]. The human immunodeficiency virus trans-activator of transduction (TAT) is a small basic CPP which contains six arginine and two lysine residues and was the first cationic peptide that demonstrated translocation properties [118]. Subsequently, many other peptides, like SynB, Penetratin, Angiopep-2, dNP2, and PepH3, were studied with relevant results [119,120].

For RMT, ligand-specific receptors such as low-density lipoprotein receptors (LDLR), transferrin receptors (TfR), lactoferrin receptors (LfR), and acetylcholine receptors (AchR) are expressed in the BBB and specialized for the transport of substances through this pathway from the blood to the brain. RMT is a promising method of drug delivery involving the endocytosis of macromolecule-sized drugs, and more recently it has been involved in NP delivery into the brain. Uptake can occur via clathrin-mediated or caveolin-mediated endocytosis, or alternatively via lipid raft internalization, a less-explored pathway [121].

NPs modified and functionalized with targeting moieties that specifically interact with BBB endothelial cells lead to an accumulation of the nanosystems, as well as the associated therapeutics, in the target site [122,123,124]. Typically, these surface-engineered DDSs follow the two above-described most popular pathways to overcome the BBB and target the brain.

Also, a different and interesting strategy to improve BBB-overcoming is based on magnetic stimulation using MNPs as active targeting moieties. Research has proven that the MNPs’ unique properties of responding to magnetic fields improve in vivo brain permeability of SPIONs and consequently increase the concentration of drugs grafted in the nanoparticulate system in the brain [125,126,127]. Moreover, magnetic heating proved evidence of the increasing of BBB permeability, indicating a substantial but reversible opening of the barrier where the hyperthermia effect of MNPs was applied [128].

The following section will describe the application of targeted nanosystems as specific AD therapeutic platforms that exclusively transpose the BBB via RMT or AMT.

### 4.2. RMT-Targeted Nanosystems

The low-density lipoprotein receptor-related protein (LRP) is a member of the LDLR family that can bind numerous ligands, including proteinases, proteinase inhibitor complexes, and certain ApoE- and lipoprotein lipase-enriched lipoproteins, mediating the cellular internalization of these ligands and their transport across the BBB [115]. In an attempt to improve the delivery and bioavailability of AD drugs to the brain, rivastigmine and tacrine, both cholinesterase inhibitors, were bonded to poly(n-butyl cyanoacrylate) (PPBCA) NPs alone and also in combination with the NPs coated with polysorbate 80, a non-ionic surfactant. These NPs were able to mimic LDL, interact with LDLR, and increase the drug concentration in rat brains 3.2-fold for rivastigmine and 4.07-fold for tacrine when using one percent of surfactant when compared to the free drug, respectively [129,130]. Also, Jose et al. [131] encapsulated bacoside-A into PLGA NPs modified with polysorbate 80 to evaluate the brain accumulation of the delivery system in rats. Bacoside-A is a plant extract that has been tested for the treatment of neurodegenerative disorders such as AD once it was reported to significantly improve the acquisition, consolidation, and retention of memory. Polysorbate 80-coated PLGA NPs were able to deliver 10-fold more bacoside-A into the brain compared to the free drug solution (23.95 ± 1.74 μg/g tissue vs. 2.56 ± 1.23 μg/g tissue), which highlighted the role of polysorbate 80-modified PLGA NPs on brain targeting and a potential treatment for AD [131]. A report suggests that polysorbate 80-coated NPs trigger temporary BBB disruption and thereby gain forced entry into the brain parenchyma [132]. However, most authors indicate that this surfactant induces the adsorption of apolipoproteins such as ApoE or ApoA-I, and that these apolipoproteins in turn interact with LDLR to trigger RMT of the conjugated NP drug system [133,134]. Polysorbate 80-NPs have been used to successfully deliver several drugs to the brain, including dalargin, kytorphin, loperamide, tubocurarine, and doxorubicin. As an alternative to this surfactant, ApoE itself can be directly and covalently functionalized to the NP surface, facilitating transcytosis [135].

Song et al. [136] constructed a biologically inspired nanostructure based on ApoE3 (an isoform of ApoE) in combination with high-density lipoprotein (ApoE3-rHDL). This assembly possessed the ability to cross the BBB, presented high binding affinity to Aβ, and in turn facilitated its clearance (Figure 7A). About 0.4% ID/g of ApoE3–rHDL reached the mouse brain 1 h after i.v. administration (Figure 7B). Moreover, a four-week daily treatment with ApoE3–rHDL decreased Aβ deposition, attenuated microgliosis, ameliorated neurologic changes, and rescued memory deficits in senescence-accelerated, P8 strain (SAMP8) mice (Figure 7C). These results indicated that ApoE3–rHDL could serve as a novel brain-targeted nanomedicine for AD therapy.

In another study, curcumin-loaded poly(butyl)cyanoacrylate (PBCA) NPs decorated with ApoE3 ligands exhibited low-density lipoprotein (LDL) receptor (ApoE3-C-PBCA)-facilitated transcytosis across the BBB and through SH-SY5Y neuroblastoma cells [137]. The inhibition of Aβ_1–42_-mediated toxicity by ApoE3-C-PBCA nanocarriers was evaluated and compared with free curcumin on SHSY5Y cells. The results indicated a reduction of 40% compared with the free drug at 100 nM Aβ of Aβ_1–42_-mediated toxicity on cells treated with the functionalized nanoparticles along with a reduction of toxic reactive oxygen species (ROS) formation [137].

Transferrin (Tf) is a protein capable of binding and carrying iron through the human body and is a naturally occurring TfR ligand [138]. TfR is highly expressed in BBB endothelial cells and is considered a mechanism that increases biological substance uptake into the brain [139]. Visser et al. [140] produced pegylated liposomes loaded with horseradish peroxidase (HRP) and tagged with transferrin (Tf) to target BBB in vitro. After 2 h of incubation of brain capillary endothelial cells (BCECs) with both Tf-modified (lipo Tf) and unmodified liposomes (lipo C) at 37 °C, the cell uptake of lipo-Tf was found to be 1–3 times higher than the lipo C.

OX26 is a monoclonal antibody (mAb) against TfR on BCECs. Pang et al. [141] developed poly(ethylene glycol)-poly(ε-caprolactone) (PEG-PCL) polymersomes (PO) conjugated with mouse-anti-rat mAb OX26 (OX26-PO) and studied their accumulation in rats’ brain tissues. OX26 density in PO was revealed to play a key role in brain accumulation. Moreover, the encapsulation of an AD therapeutic peptide, NC-1900, within OX26-PO also presented significant learning and memory improvements in an AD rat model in a water maze task. Another study performed by Loureiro et al. [142] showed that pegylated liposomes functionalized with OX26 and an anti-amyloid beta peptide monoclonal antibody (19B8) improved cellular uptake of the nanocarrier in porcine BCECs about eight-fold compared to the control group. Additionally, the nanocarrier ability to cross the BBB was established by in vivo studies in wild-type rats and it was demonstrated that the dual antibody-decorated immunoliposomes could traverse the barrier.

Nevertheless, the use of large proteins, such as Tf and OX26, can easily cause challenges in formulation stability and immunological response. In this sense, short peptides are represented as good candidates for the overcoming of the BBB. Peptides are used as targeting delivery systems due to their high specificity, low cytotoxicity, and low immunological response. The B6 peptide (CGHKAKGPRK) was discovered by phase display as a substitute for Tf with a high affinity for TfR. The conjugation of the B6 peptide to the surface of the PEG–PLA block copolymer NPs (B6-NP) exhibited higher internalization in BCECs, which is corroborated by the enhanced brain accumulation in BALB/c nude mice when compared with unmodified NPs. Moreover, the administration of the B6-NP-encapsulated neuroprotective peptide, NAPVSIPQ (NAP), to AD mouse models revealed excellent amelioration in learning impairments, cholinergic disruption, and loss of hippocampal neurons even at a lower dose (0.02 μg/day). In contrast, the free NAP at concentrations up to 0.08 μg/day failed to produce any significant enhancement [143].

Lactoferrin (Lf) has a molecular weight of 80 kDa and is a naturally occurring iron-binding glycoprotein of the Tf family. Lactoferrin receptor (LfR) is highly expressed on the apical surface of respiratory epithelial cells, as well as in brain endothelial cells and neurons, and is particularly overexpressed in capillaries and neurons associated with age-related neurodegenerative diseases, including AD, Parkinson’s disease (PD), and amyotrophic lateral sclerosis (ALS) [144]. Thus, Lf might be a promising brain-targeting ligand for drug delivery systems for CNS diseases. Huperzine A (HupA) is a reversible inhibitor of acetylcholinesterase (AChE) which enhances memory in behavioral animal models. Meng et al. [145] built co-modified Lf and N-trimethylated chitosan (TMC) HupA-loaded PLGA NPs (Lf-TMC NPs) for the efficient intranasal delivery of HupA to the brain for AD treatment. Qualitative and quantitative cellular uptake experiments indicated that the accumulation of Lf-TMC NPs was higher than nontargeted analogs in 16HBE and SH-SY5Y cells. In vivo imaging results showed that the targeted nanosystem exhibited a higher fluorescence intensity in the brain and a longer residence time compared to the nontargeted NPs. After intranasal administration, Lf-TMC NPs facilitated the distribution of HupA in the brain, and the values of the drug-targeting index in the mouse olfactory bulb, cerebrum (with hippocampus removal), cerebellum, and hippocampus showed very significant differences from those of the nontargeted group.

Although the mentioned examples are focused on RMT transcytosis with ligand-specific receptors, such as LDLR, TfR, and LfR, other methods have been employed to allow BBB penetration of nanoparticle-based Alzheimer’s therapeutics via the RMT. Notably, the research conducted by Liu et al. has explored an alternative approach where they developed a dual-targeted magnetic mesoporous silica nanoparticle coated with hyaluronic acid (HA), a non-immunogenic glycosaminoglycan, that is recognized by the CD44 surface receptor. As a dual-targeted Aβ clearance system, the HA-coated magnetic mesoporous silica nanoparticle was further functionalized with an anti-Aβ42-targeting antibody 1F12 (HA-MMSN-1F12) to capture Aβ42 peptides. In vivo experiments reveal that the group was able to produce non-toxic NPs that accumulated in the brain and degraded Aβ42 aggregates, consequently reducing neuroinflammation and improving memory deficits [146].

### 4.3. AMT-Targeted Nanosystems

Wen et al. [147] described TAT-functionalized magnetic PLGA-lipid NPs (MPLs) formed by PLGA, L-α-phosphatidylethanolamine (DOPE), DSPE-PEG-NH2, and SPIONs. The TAT-MPLs were designed to encapsulate antioxidant and anti-inflammatory properties, like the compounds hesperidin (HES), naringin (NAR), and glutathione (GSH), and also to target the brain by magnetic guidance and TAT conjugation. TAT conjugation of MPLs could significantly enhance the cellular delivery and the therapeutic efficacy of the compounds in immortalized mouse brain endothelial (bEnd.3) cells by penetrating the cell membrane compared with non-conjugated MPLs.

Furthermore, Zhao et al. [148] developed a silica (SiO_2_)-coated magnetic nanoparticle-based carrier (SiO_2_@Fe_3_O_4_) conjugated to the TAT peptide (SiO_2_@Fe_3_O_4_-TAT) to evaluate its ability to cross the BBB. SiO_2_@Fe_3_O_4_-TAT NPs added to the apical chamber of the in vitro BBB model were found in U251 glioma cells co-cultured at the bottom of the Transwell, indicating the cellular uptake by these cells after crossing human cardiac microvascular endothelial (hCMEC) cells. The conjugation with TAT along with the applied magnetic field in the in vitro model contributed to a synergistic effect for cellular internalization and permeability across the barrier, suggesting that the nanosystem could penetrate the BBB via transcytosis and magnetically mediated dragging. Although the authors have achieved an important in vitro accomplishment using the TAT peptide as a vehicle for NP delivery, it is important to mention that the in vivo brain targeting was not investigated in either of the studies.

Some categories of opioid peptides have been shown to penetrate the BBB and to be able to cross this barrier to a higher extent when glycosylated. Thus, an opioid peptide able to enter the brain was modified to eliminate the opioid activity [149]. The new simil-opioid peptide synthetized was then glycosylated (H_2_N-Gly-l-Phe-d-Thr-Gly-l-Phe-l-Leu-l-Ser[*O*-b-d-glucose]-CONH_2_; g7) and conjugated to the PLGA polymer to obtain, through a nanoprecipitation procedure, an engineered NP (g7-NPs) able to enter the brain [150,151]. The investigators performed comparison experiments (pharmacological and biodistribution-based) to predict the mechanism of NP passage through the BBB, based on the ability/inability of g7-NPs and random-g7-NPs to deliver drugs across the BBB. Loperamide (LOP), a known model drug unable to cross the BBB, was encapsulated within the NPs, and only LOP delivered to the brain with g7-NPs created high central analgesia, corresponding to 14% of the injected dose in male albino Wistar Hannover rats. Without damaging the BBB, these NPs were hypothesized to use a membrane–membrane interaction and macropinocytosis-like mechanisms as the pathway for BBB crossing. Due to the great amount of NP localization in the brain, the authors reject the hypothesis of a specific receptor mediating the BBB crossing of g7-NPs [150].

To counteract the imbalance of zinc caused by Aβ plaques, Vilella et al. [152] designed a PLGA system modified with the g7 glycopeptide for zinc brain delivery (g7-NPs-Zn). WT and APP23 mice were treated with g7-NPs-Zn to study the action of increased brain zinc levels on AD pathology. The system was able to reach the brain, and through atomic absorption spectrometry (AAS) it was observed that g7-NPs-Zn led to an increase of zinc levels in APP23 mice brains. This effect promoted both reductions of Aβ plaque formation and pro-inflammatory cytokine levels while contributing to the stabilization of synapse density [152,153].

Trimethylated chitosan (TMC) is a permanently quaternized chitosan (CS) derivative that is positively charged under physiological conditions. As a cationic ligand, TMC facilitates the brain transport of NPs through the AMT pathway to enhance drug delivery into the CNS. Coumarin-6-loaded PLGA-NP (Figure 8A) and TMC-modified PLGA-NP (Figure 8B) were injected into the caudal vein of mice, and fluorescent microscopy of brain sections showed a higher accumulation of TMC-modified PLGA-NP in the cortex, third ventricle, and choroid plexus epithelium, while no brain uptake of unmodified NPs was observed [154]. Coenzyme Q_10_ was chosen as the AD model drug for the evaluation of the neuroprotective effects of TMC-modified PLGA NPs in APP/PS1 transgenic mice. Behavioral testing showed that the injection of coenzyme Q_10_-loaded TMC-modified PLGA-NP greatly improved memory impairment, restoring it to a normal level, and decreased the number of senile plaques when compared with unmodified NPs (Figure 8C,D). Thus, TMC modification enabled NPs to transport across the BBB and effectively deliver the drug into the brain [154].

Despite the fact that vesicles formed during AMT have a greater capacity (i.e., they accommodate larger macromolecules) than those formed during RMT, drawbacks associated with AMT include its lack of selectivity (adsorption may occur not only at the BBB, but also in the blood vessels of other organs) [116], which could limit its application in drug delivery systems for AD therapy.

An outlook of different examples of nano-DDSs described in the literature for AD therapy and diagnosis are listed in Table 1.

## 5. Conclusions

In the context of AD, the development of targeted nanosystems holds great promise in overcoming the challenge of delivering therapeutic molecules across the BBB for an effective treatment of the disease. Since the BBB acts as a formidable obstacle to many drugs’ entrances into the brain, researchers have devised targeted nanosystems that utilize RMT and AMT transporters as potential pathways for transporting NPs to the specific affected area in the brain. The integration of targeted nanosystems with advanced imaging techniques, such as MRI, represents a promising approach to address the challenge of drug delivery across the BBB while allowing AD diagnosis and therapeutic monitoring.

Iron oxide nanoparticles such as SPIONs have demonstrated their effectiveness as MRI contrast agents in visualizing and detecting AD-related pathological features, including amyloid plaques, in both in vitro and in vivo settings. Moreover, their functionalization with specific ligands, such as Aβ peptides or targeting moieties, has enhanced their affinity for those AD-related biomarkers and improved their ability to cross the BBB.

Moreover, the biocompatibility and safety of SPIONs have been demonstrated with their nontoxic nature and biological tolerance. After injection, these NPs can be integrated into human natural processes of iron metabolism, further supporting their potential clinical application as diagnostic tools for AD, and ensuring safety during in vivo use. The ability to modify the surface charge and hydrophobic character of SPIONs is essential for their long-term diagnostic and therapeutic applications. These coatings prevent particle aggregation, degradation, and structural alterations, preserving the NPs’ magnetic and structural properties over time. Also, some coatings can respond to specific stimuli, such as changes in pH, temperature, or enzymes, triggering controlled drug release at the targeted site. This feature enhances the efficiency and efficacy of treatments, ensuring that therapeutic agents are delivered precisely where they are needed.

Overall, the integration of diagnostic and therapeutic functions into a single system allows for a comprehensive approach to AD theranostics. Theranostics’ NPs offer the potential for personalized medicine, where diagnosis, targeted drug delivery, and real-time monitoring of treatment response can be achieved simultaneously. This could revolutionize the way AD is diagnosed and managed, potentially leading to more effective and tailored treatments for patients. Nevertheless, future research should focus on enhancing the selectivity of delivery systems, improving specificity in targeting ligands to minimize off-target effects, and increasing drug delivery efficiency. Additionally, exploring the use of external stimuli, such as magnetic fields, to trigger controlled drug release from nanosystems can further enhance their therapeutic potential.

Despite the revised work in this study, researchers are continually working to address limitations associated with the in vivo use of magnetic nanoparticles. One of the main concerns is their potential for toxicity associated with the nanoparticle features that may affect their biocompatibility, biodistribution, and clearance. Addressing these challenges requires personalized nanoparticle synthesis and surface modifications to enhance safety and reduce size-dependent effects. Scaling up production, conducting rigorous clinical trials, and evaluating safety profiles are necessary for successful human implementation.

In summary, targeted multifunctional magnetic nanosystems hold immense promise for advancing AD theranostics. These specialized nanoparticles offer a multifaceted approach to tackling the challenges associated with the ailment, ultimately aimed at enhancing the quality of life of patients in the fight against AD.

## Figures and Tables

**Figure 1 pharmaceutics-15-02316-f001:**
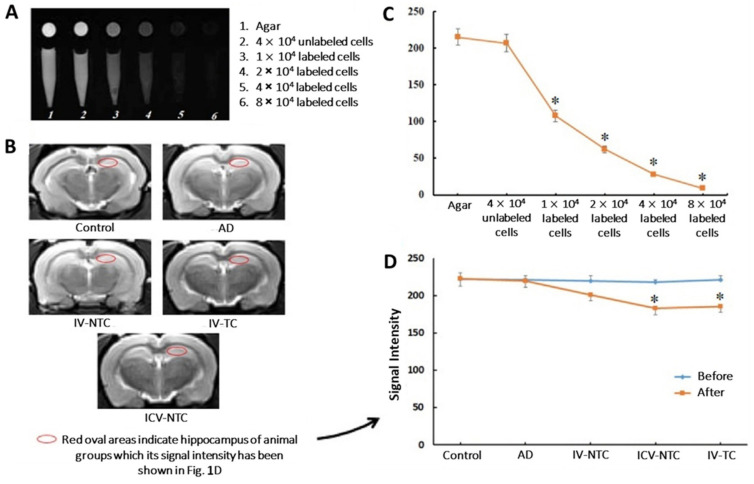
In vitro and in vivo MRI evaluation. (**A**) Coronal and transverse MRI images from microtubes containing different concentrations of SPION-labeled cells under 1.5-Tesla. (**B**) Coronal T2 Weighted Turbo Spin Echo (T2W-TSE) sequence from the rat brain of different groups after cell treatment using 3-Tesla MRI scanner. (**C**) Signal intensity diagram of MRI images (* *p* ≤ 0.05 compared with agar and unlabeled cells). (**D**) Signal intensity of the hippocampus region in MRI images in different groups before and after cell treatment. (* *p* ≤ 0.05 compared with control and AD). Control: Wild-type (WT) rat group; AD: AD model group; IV-NTC: AD rats treated intravenously (IV) with non-targeted cells; IV-TC: AD rats treated IV with targeted cells; ICV-NTC: AD rats treated with intracerebroventricular (ICV) non-targeted cells. (Adapted with permission from [71]. Copyright © 2023, Elsevier, Amsterdam, The Netherlands).

**Figure 2 pharmaceutics-15-02316-f002:**
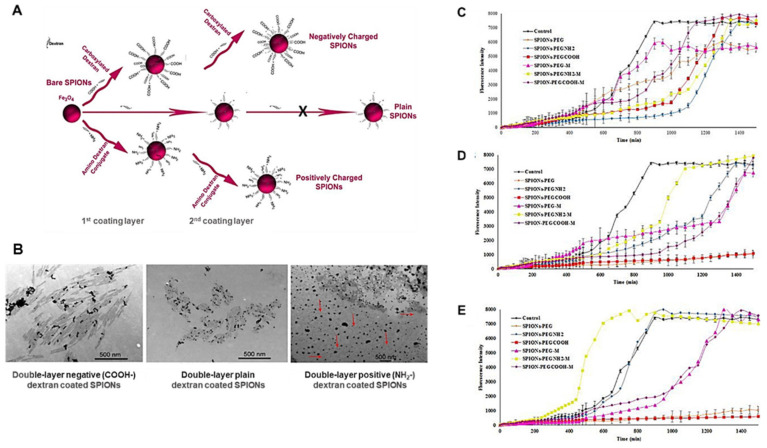
(**A**) Scheme of SPION preparation with single- and double-negative or positive dextran coatings. (**B**) Transmission electron microscopy (TEM) images of Aβ fibrils with double-layer negative, plain, and positive dextran-coated SPIONs at two different magnifications. (Adapted with permission from [72]. Copyright © 2023, American Chemical Society, Washington, DC, USA). Influence of magnetic field application in the kinetics of Aβ aggregation with negative, positive, and plain PEG-coated SPIONs at different concentrations including (**C**) 40 μg/mL, (**D**) 80 μg/mL, and (**E**) 100 μg/mL (Adapted with permission from [73]. Copyright © 2023, Elsevier, Amsterdam, The Netherlands).

**Figure 3 pharmaceutics-15-02316-f003:**
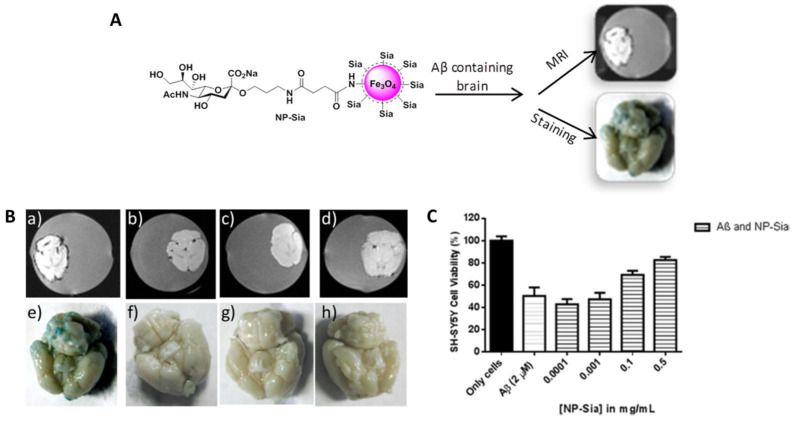
(**A**) Schematic illustration of NP-Sia formulation and their dual approach to detect Aβ deposits. (**B**) T2* weighted MRI (**a**–**d**) and Prussian blue staining (**e**–**h**) images of NP-Sia incubation in mice brains under different conditions. (**C**) Neuroprotection effect in SH-SY5Y cells after addition of NP-Sia. (Adapted with permission from [90]. Copyright © 2023, American Chemical Society, Washington, DC, USA).

**Figure 4 pharmaceutics-15-02316-f004:**
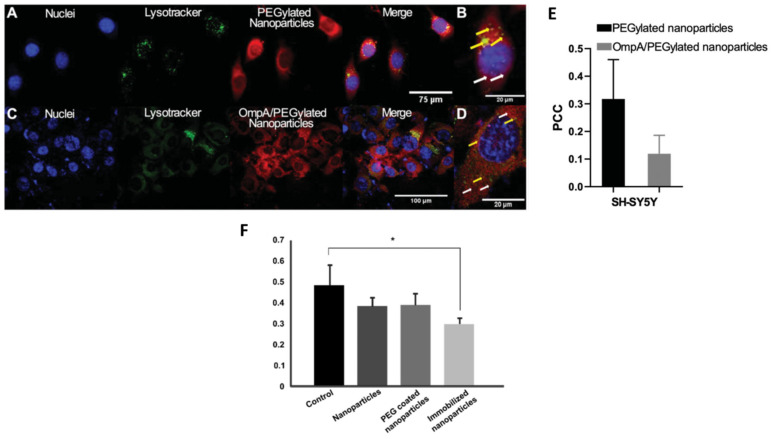
(**A**–**D**) Confocal microscopy images of endosomal escape of PEGylated- and OmpA/PEGylated-SPIONs in SH-SY5Y cells. (**E**) Pearson’s correlation coefficient (PCC), for both PEGylated-and OmpA/PEGylated-SPIONs in SH-SY5Y cells, indicates endosomal escape increase in case of particles conjugated with the translocator molecule OmpA. (**F**) Relative BACE1 expression in different treatment conditions. (* *p* ≤ 0.05 compared with control). (Adapted with permission from [99]. Copyright © 2023, Taylor & Francis Ltd., London, UK).

**Figure 5 pharmaceutics-15-02316-f005:**
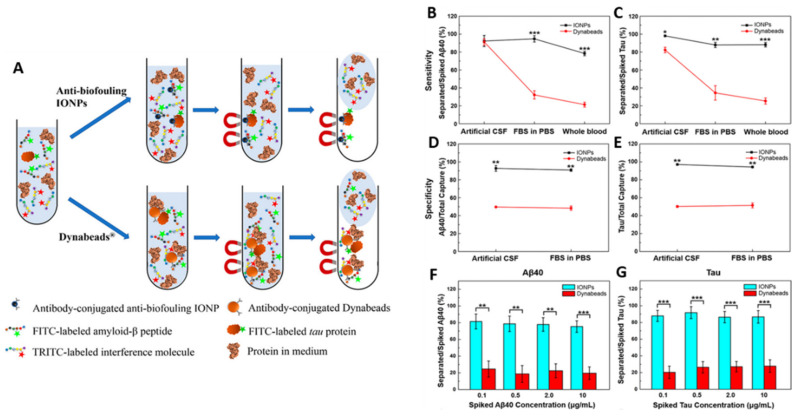
(**A**) Schematic illustration of immunomagnetic separation of Aβ and tau proteins in CSF or serum by antibody-conjugated antibiofouling SPIONs or Dynabeads^®^. Sensitivity comparison between SPIONs and Dynabeads^®^ on Aβ40 (**B**) and tau (**C**) proteins separation. Specificity of SPIONs and Dynabeads^®^ on separation of Aβ40 (**D**) and tau (**E**) proteins. Separation of Aβ40 (**F**) and tau (**G**) proteins from human blood with the different magnetic separating agents. (* *p* < 0.05, ** *p* < 0.01, *** *p* < 0.001) (Adapted with permission from [100]. Copyright © 2023, American Chemical Society, Washington, DC, USA).

**Figure 6 pharmaceutics-15-02316-f006:**
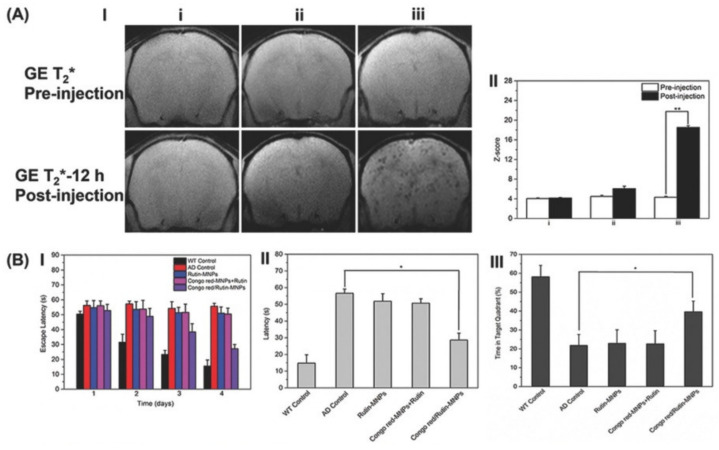
(**A**-**I**) In vivo MRI representation of brains in WT mice before and after injection of Congo red/Rutin-MNPs (**i**) and AD mice before and after injection of Rutin-MNPs (**ii**) and Congo red/Rutin-MNPs (**iii**). (**A**-**II**) Z-scores from the MRI data among WT mice before and after injection of Congo red/Rutin-MNPs (**i**) and AD mice before and after injection of Rutin-MNPs (**ii**) and Congo red/Rutin-MNPs (**iii**). GE: gradient-echo, a type of pulse sequence during image acquisition. (**B**) Escape latencies (**B-I**) and latency during the memory test in Morris water maze (MWM) probe trial without a platform (**B-II**). Time in the target quadrant in MWM probe trial with a platform (**B-III**). All tests were performed in WT control mice, AD control mice, and AD mice treated with Rutin-MNPs, Congo red-MNPs + Rutin, and Congo red/Rutin-MNPs. (* *p* < 0.05, ** *p* < 0.01). (Adapted with permission from [107]. Copyright © 2023, John Wiley and Sons, Hoboken, NJ, USA).

**Figure 7 pharmaceutics-15-02316-f007:**
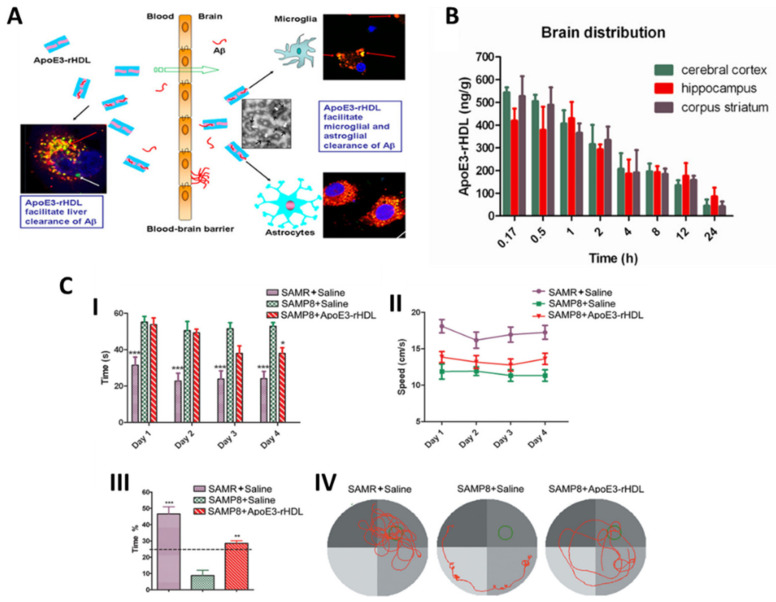
(**A**) Schematic representation of ApoE3–rHDL action for Aβ clearance from microglia and astroglia after BBB permeation. (**B**) Brain distribution of ^125^I-ApoE3–rHDL after intravenous administration. (**C**) ApoE3–rHDL rescued memory deficits in SAMP8 mice. (**I**) Escape latency, (**II**) swimming speed, (**III**) time spent in the quadrant where the escape platform is located, and (**IV**) representative swimming path. (* *p* < 0.05, ** *p* < 0.01, *** *p* < 0.001 significantly different with those of the saline treated SAMP8 mice.) (Adapted with permission from [136] Copyright © 2023, American Chemical Society, Washington, DC, USA).

**Figure 8 pharmaceutics-15-02316-f008:**
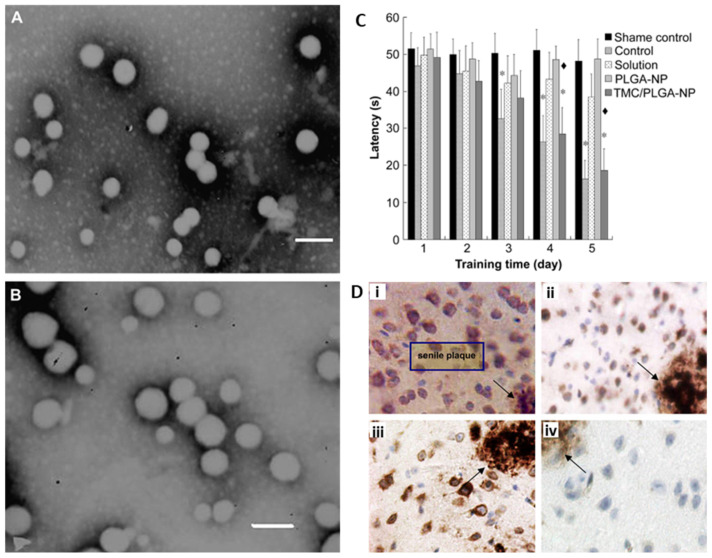
TEM of PLGA-NP (**A**) and TMC-modified PLGA–NP (**B**). Scale bars are 200 nm. (**C**) Latency during memory test in MWM in APP/PS1 transgenic mice for different conditions. (**D**) Effects of different formulations in cells around senile plaques in APP/PS1 transgenic mice after daily application of (**i**) saline, (**ii**) coenzyme Q_10_ solution, (**iii**) coenzyme Q_10_-loaded PLGA-NPs, and (**iv**) coenzyme Q10-loaded TMC-modified PLGA-NPs. (* *p* < 0.05, significantly different from shame control; ^♦^
*p* < 0.01, significantly different from PLGA–NP) (Adapted with permission from [154] Copyright © 2023, Elsevier, Amsterdam, The Netherlands).

**Table 1 pharmaceutics-15-02316-t001:** Representative studies using different types of NPs as drug delivery systems for Alzheimer’s therapy and diagnosis.

Nanoformulation	Targeting Moiety	Therapeutic/Diagnostic Agent	Size (Zeta Potential)	Animal Model/Cell Line/Brain Samples	Administration Route	In Vitro/In Vivo Results (Main Outcomes)	Refs
PLGA PEG NPs	CRT	Nec-1s	90 ± 15.2 nm (Not available)	APP/PS1 transgenic mice	Intraperitoneal injection	▪ CRT and CD47 conjugation improved BBB permeation and blood circulation, respectively. ▪ Overcoming of Nec-1s, a RIPK1 inhibitor, which has poor water solubility and low brain bioavailability itself. ▪ Decreased levels of oxidative stress, Aβ plaques, and inflammatory cytokines resulted in the improvements on neuron survival and in the attenuation of cognitive deficits.	[155]
PLGA PEG NPs	-	Memantine	156.6 ± 0.5 nm (−22.4 mV)	APP/PS1 transgenic mice	Oral	▪ The therapeutic effectiveness of memantine was improved. ▪ NPs reached the brain and showed a prolonged therapeutic effect in comparison to free memantine. ▪ Improvement of memory impairments and reduction of Aβ plaques formation.	[156]
Polysorbate 80-coated PLGA NPs	-	Donepezil	89.7 ± 6.4 nm (−36.0 ± 1.0 mV)	Sprague–Dawley rats	Intravenous injection	▪ Development of a long-term donepezil drug delivery system. ▪ Donepezil accumulated in the brain in a higher extension when loaded in P80-coated PLGA-NPs in comparison to its free form.	[157]
PLGA (DSPE-PEG-T807 + red blood cell membrane) NPs	T807	Curcumin	170.7 ± 0.7 nm (+8.9 ± 0.4 mV)	Sprague–Dawley rats (okadaic acid treated)	Intravenous injection	▪ Improved curcumin delivery into the brain. ▪ T807-targeting ligand improved BBB permeation and brain accumulation, while red blood cell membrane coating promoted blood circulation and biocompatibility. ▪ Suppressed AD progression in vivo by the reduction of intracellular p-tau levels, oxidative stress, and neuronal cell death.	[158]
PLGA PEG NPs	CRT	Aβ generation inhibitor S1 and Curcumin	139.8 ± 15.2 nm (−25.7 mV)	APP/PS1 transgenic mice	Intraperitoneal injection	▪ Improved accumulation in the brain of both molecules. ▪ Improved spatial memory and recognition. ▪ Decreased levels of Aβ plaques, ROS, tumor necrosis factor-alpha, and interleukin-6. ▪ Enhanced super oxide dismutase (SOD) activity and synapse number in the AD mouse brains.	[159]
PLGA-Chitosan NPs	K16ApoE	IgG4.1 and Curcumin	235 ± 10 nm (+4.9 ± 0.1 mV)	Tg2576 transgenic mice	Intravenous injection	▪ Improving BBB targeting and brain accumulation of IgG4.1 and curcumin. ▪ Improved tissue distribution and brain uptake. ▪ Improved capacity to detect Aβ plaques in brain.	[160]
PEG-Liposomes (DPS-PEG_2000_-Anti-TfR mAbs)	OX26 and RI7217	Curcumin	153 ± 11 nm (−7.5 ± 1.2 mV)	Human samples from the superior temporal gyrus (Brodmann area 22) (brain donation from AD patients) hCMEC/D3 cells	-	▪ Nanoliposome (NL) construct demonstrated to strongly label Aβ deposits in *post mortem* tissues. ▪ NLs effectively inhibited Aβ_1–42_ aggregation. ▪ Cellular uptake studies demonstrated that curcumin derivate NLs were able to internalize cells with a slight reduction in comparison with non-curcumin derivate NLs.	[161]
PEG-Liposomes (Sm/Chol/mal-PEG)	Phosphatidic acid and ApoE-derived peptide	-	123 ± 3 nm (−15.2 ± 1.1 mV)	Balb/c mice hCMEC/D3 cells	Intravenous injection (for in vivo)	▪ Studies showed that bi-functionalized liposomes strongly bind Aβ deposits (*K*_D_ = 0.6 μM), inhibit protein aggregation (70% inhibition after 72 h), and trigger the disaggregation of preformed aggregates (60% decrease after 120 h incubation). ▪ Bi-functionalization enhanced liposome passage across the BBB either in vitro or in vivo in healthy mice.	[162]
PEG-Liposomes (DOPE/DOTAP/Chol/DSPE-PEG_2000_)	RVG, Pen and MAN	pApoE2 and chitosan	172 ± 3.09 nm (+19.0 ± 0.9 mV)	C57BL/6 mice bEnd.3 cells	Intravenous injection (for in vivo)	▪ RVGMAN and PenMAN liposomes encapsulating ApoE2/chitosan complex significantly improved transport and transfection of ApoE2 gene through BBB. ▪ Dual-functionalized liposomes prevented ApoE2 from endonuclease digestion and were effective in brain-targeting and expression of genetic cargo in brain cells. ▪ ApoE2 expression levels significantly increased in C57BL/6 mouse brain compared to formulation controls.	[163]
PEG-Liposomes (Sm/Chol/mal-PEG-azido)	TAT	Curcumin	196.5 ± 3.2 nm (−12.94 ± 0.94 mV)	hCMEC/D3 cells	-	▪ TAT promoted a higher internalization of NLs in hCMEC/D3 cells compared with bare formulation. ▪ TAT functionalization increased the permeability of curcumin-NLs across the in vitro BBB model. The similar permeabilities of curcumin derivative and [^3^H]-sphingomyelin suggested that NLs were transported intact.	[164]
Gd-coated Chitosan NPs	IgG4.1	Curcumin and dexamethasone (therapeutics) Gadolinium (diagnostic)	145 ± 5.4 nm (+7.7 ± 0.4 mV) for curcumin-NPs 157.6 ± 3.4 nm (+4.5 ± 0.5 mV) for dexamethasone-NPs	B6/SJL mice Tg2576 transgenic mice	Intravenous injection	▪ IgG4.1 functionalization improved the targeting of NPs to cerebrovascular amyloid (CVA) deposits. ▪ The NPs complex effectively migrated from the blood flow to the vascular wall and demonstrated excellent distribution in brain vasculature. ▪ Accumulation of therapeutic agents to reduce cerebrovascular inflammation associated with cerebral amyloid angiopathy.	[165]
Magnetic (iron oxide) NPs	Heparin	-	68 nm (−53.3 mV)	SH-SY5Y cells	-	▪ High affinity for Aβ fibrils association and protection of neuronal cells against Aβ toxicity.	[166]
Magnetic (iron oxide) NPs	AβPP	-	9.5 ± 1.0 nm (TEM) (−42 mV)	APP/PS1 transgenic mice	Intravenous injection	▪ The conspicuity of the plaques increased from an average Z-score of 5.1 ± 0.5 to 8.3 ± 0.2 when the plaque contrast-to-noise ratio was compared in control AD mice with AD mice treated with magnetic NPs, indicating that anti-AβPP-conjugated NPs crossed the BBB. ▪ The number of MRI-visible plaques per brain increased from 347 ± 45 in the control AD mice to 668 ± 86 in the magnetic NP-treated mice.	[82]
PEG-coated magnetic (iron oxide) NPs	NU4	-	30 nm (−40 mV)	5xFAD mice	Intranasal	▪ NU4-conjugated magnetic NPs were able to bind (24–48)-unit Aβ oligomers in AD mice brains. ▪ The nanosystem is both specific and sensitive and can distinguish AD brain tissue from non-demented controls by MRI in vitro. ▪ In vivo, the probe reaches the brain, distributes the intended targets within 4 h, as well as shows significant clearance from the brain within four days after introduction.	[167]
Curcumin-conjugated magnetic (iron oxide) NPs (coated with PEG-PLA block copolymer and PVP polymer)	-	Curcumin	93.4 ± 3.0 nm (−0.38 ± 0.13 mV)	Tg2576 transgenic mice	Intravenous injection	▪ Curcumin-magnetic NPs can penetrate the BBB of Tg2576 AD model and effectively bind amyloid plaques. ▪ T2* ex vivo MRI reveals more dark spots in AD mice brains than in control mice, which are aligned with amyloid plaques on immunohistochemically stained sections.	[69]

List of abbreviations: Aβ: amyloid beta; AβPP: anti-Aβ protein precursor; AD: Alzheimer’s disease; ApoE: *Apolipoprotein E*; BBB: blood–brain barrier; Chol: cholesterol; CRT: iron-mimic cyclic peptide CRTIGPSVC; DOPE: dioleoyl phosphatidylethanolamine; DOTAP: 1,2-dioleoyl-3-trimethylammoniumpropane; DSPE: 1, 2-Distearoyl-sn-glycero-3-phosphoethanolamine; mAbs: monoclonal antibodies; Gd: gadolinium; IgG4.1: monoclonal antibody against human fibrillar Aβ42; K16ApoE: 16 lysine (K) residue-linked LRP-binding amino acid segment of ApoE; Mal: maleimide; MAN: mannose; MRI: magnetic resonance imaging; Nec-1s: RIPK1 inhibitor; NL: nanoliposome; NPs: nanoparticles; NU4: Aβ oligomer-specific monoclonal antibody NU4; OX26: anti-TfR monoclonal antibody OX26; pApoE2: ApoE2 encoding plasmid DNA; PEG: poly(ethylene) glycol; Pen: penetration; PLA: poly(lactic acid); PLGA: poly(lactic-co-glycolic acid); PVP: polyvinylpyrrolidone; RIPK1: receptor-interacting serine/threonine-protein kinase 1; Rl7217: anti-TfR monoclonal antibody Rl7217; ROS: reactive oxygen species; RVG: rabies virus glycoprotein peptide; Sm: sphingomyelin; SOD: super oxide dismutase; T807: ^18^F-T807 Tau positron emission tomography tracer; TAT: virus trans-activator of transduction peptide; TfR: transferrin receptor.

## Data Availability

Not applicable.

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
