# Peer review of "A Promising Approach: Magnetic Nanosystems for Alzheimer’s Disease Theranostics"

_pharmaceutics, 2023, doi:10.3390/pharmaceutics15092316_

Round 1
Reviewer 1 Report
The authors have conducted a comprehensive review of the literature pertaining to multifunctional magnetic nanoparticles (MNPs) and their evolving role as a dual-faceted nanotechnological platform for both diagnosing and treating neurodegenerative disorders. This review article demonstrates a commendable level of articulation and serves as a valuable source of information within the concerned research domain. The present reviewer contends that, subject to minor typographical revisions, this manuscript is suitable for publication. The included table significantly enhances the overall quality of the review.
Several constructive comments are presented below:
-
Line 39: The assertion that AD ranks as the second most prevalent disorder prompts an inquiry into the identity of the most predominant ailment within this context. It is suggested that the authors consider augmenting this information to afford a clearer perspective.
-
Line 771: An error appears to have been encountered. Further clarification or correction is warranted at this juncture.
-
It has come to the reviewer's attention that certain sections, including "Patents" and "Acknowledgments," remain incomplete or unwritten. The authors are encouraged to address this discrepancy for the sake of completeness.
-
Figure 7: The acronym "GE" utilized within the context of this figure remains ambiguous. The authors are encouraged to provide an elucidation of its intended meaning to ensure comprehensive reader understanding.
In conclusion, the authors have effectively surveyed the expansive landscape of multifunctional MNPs in the context of their dual role in diagnosing and treating neurodegenerative disorders. This review is distinguished by its clarity and informative content. Pending minor revisions, this reviewer recommends its publication for the benefit of the scientific community.
Author Response
Dear Krongkarn Watakulsin,
Thank you for your email and editorial decision on 27/08/2023 inviting us to submit a revised manuscript that addresses all the points raised by the reviewers.
We were pleased to see that reviewers consider that the work contributes to the field, is well organized and comprehensively described, deserving publication in Pharmaceutics. The authors would like to thank the reviewers for the helpful and highly relevant comments to improve the manuscript.
Please find below a point-by-point response to the reviewers’ comments.
REVIEWER 1
Major comments:
- Line 39: The assertion that AD ranks as the second most prevalent disorder prompts an inquiry into the identity of the most predominant ailment within this context. It is suggested that the authors consider augmenting this information to afford a clearer perspective.
Response: This is a valid suggestion made by the reviewer so, to clarify, we improved the sentence, as follows:
Line 39: “With around 70% of dementia cases worldwide, AD is one of the most prevalent neurodegenerative disorders. AD is a chronic and progressive brain disease that leads to deterioration of cognitive function, most commonly impaired memory, and changes in thinking and behavior. It is predicted that at the current rate, 1 in 85 persons worldwide will be living with AD by 2050, highlighting the substantial impact of AD on individuals and society as a whole [1, 4]).” The references for this paragraph were also updated.
[1] C. Ding et al., “Global, regional, and national burden and attributable risk factors of neurological disorders: The Global Burden of Disease study 1990–2019,” Front. Public Heal., vol. 10, 2022, doi: 10.3389/fpubh.2022.952161.
[4] D. Furtado, M. Björnmalm, S. Ayton, A. I. Bush, K. Kempe, and F. Caruso, “Overcoming the Blood–Brain Barrier: The Role of Nanomaterials in Treating Neurological Diseases,” Adv. Mater., vol. 30, no. 46, 2018, doi: 10.1002/adma.201801362.
- Line 771: An error appears to have been encountered. Further clarification or correction is warranted at this juncture.
Response: We thank to the reviewer for noticing this error. The error was solved and in line 771 we mean to refer to Table 1, as follows: “An outlook of different examples of nano-DDS described in the literature for AD therapy and diagnosis are listed in Table 1.”
- It has come to the reviewer's attention that certain sections, including "Patents" and "Acknowledgments," remain incomplete or unwritten. The authors are encouraged to address this discrepancy for the sake of completeness.
Response: We acknowledge that "Patents" and "Acknowledgments" sections should be included. However, as a review article, there is no technical support or donations (as suggested to be appointed by the journal in this section) to address to. The funding for this work is already mentioned in the “Funding” section. Also, there are no patents involved in the conceptualization of this review article. The information “Not applicable” was added to these sections.
- Figure 7: The acronym "GE" utilized within the context of this figure remains ambiguous. The authors are encouraged to provide an elucidation of its intended meaning to ensure comprehensive reader understanding.
Response: By mistake the meaning of the acronym “GE” was excluded. We have now added the designation “GE” means gradient-echo (GE), applicated in the context of MRI (Magnetic Resonance Imaging) and T2-weighted imaging, which is a type of pulse sequence during image acquisition. For a better reader comprehension, the acronym “GE” was included and written in Figure 7 caption.
We addressed all minor comments as highlighted in the paper (track changes).

Reviewer 2 Report
This manuscript was well written and gave adequate elaboration, while the title would be better after comfine the bounadary into Maganetic nanotechnology.
Author Response
Dear Krongkarn Watakulsin,
Thank you for your email and editorial decision on 27/08/2023 inviting us to submit a revised manuscript that addresses all the points raised by the reviewers.
We were pleased to see that reviewers consider that the work contributes to the field, is well organized and comprehensively described, deserving publication in Pharmaceutics. The authors would like to thank the reviewers for the helpful and highly relevant comments to improve the manuscript.
Please find below a point-by-point response to the reviewers’ comments.
REVIEWER 2
This manuscript was well written and gave adequate elaboration, while the title would be better after comfine the bounadary into Maganetic nanotechnology.
Response:
"A promising approach: Magnetic nanosystems for Alzheimer's disease theranostics"

Reviewer 3 Report
This manuscript reviews the utilization of magnetic nanoparticles for the diagnosis and treatment of Alzheimer's disease. The authors put forward several aspects worthy of attention in future research on magnetic nanoparticles (MNPs) focused nanotechnology. In fact, very few MNPs are in clinical trials for AD. However, the logic of this review is disordered and the content is not comprehensive due to such an extremely broad topic. The current development of nanomaterials focused on MNPs for current AD theranostics was summarized insufficiently and the discussion is not detailed. In the section of “2. Alzheimer’s disease diagnosis”, “Liquid biopsy techniques involving cerebrospinal fluid (CSF) and blood biomarkers have been included in the diagnostic criteria for Alzheimer's disease, becoming a highly studied area.” should also be mentioned. What are the potential drawbacks of a magnetic nanoparticle focused AD theranostics strategy? It is important to understand and overcome this before proceeding to clinical trials. The author should discuss this in detail in the conclusions and opinions section. The reference list does not adequately cover the latest relevant representative literature. And lacks a constructive, comprehensive and critical opinion in conclusions and perspectives.
no
Author Response
Dear Krongkarn Watakulsin,
Thank you for your email and editorial decision on 27/08/2023 inviting us to submit a revised manuscript that addresses all the points raised by the reviewers.
We were pleased to see that reviewers consider that the work contributes to the field, is well organized and comprehensively described, deserving publication in Pharmaceutics. The authors would like to thank the reviewers for the helpful and highly relevant comments to improve the manuscript.
Please find below a point-by-point response to the reviewers’ comments.
REVIEWER 3
This manuscript reviews the utilization of magnetic nanoparticles for the diagnosis and treatment of Alzheimer's disease. The authors put forward several aspects worthy of attention in future research on magnetic nanoparticles (MNPs) focused nanotechnology. In fact, very few MNPs are in clinical trials for AD. However, the logic of this review is disordered and the content is not comprehensive due to such an extremely broad topic. The current development of nanomaterials focused on MNPs for current AD theranostics was summarized insufficiently and the discussion is not detailed. In the section of “2. Alzheimer’s disease diagnosis”, “Liquid biopsy techniques involving cerebrospinal fluid (CSF) and blood biomarkers have been included in the diagnostic criteria for Alzheimer's disease, becoming a highly studied area.” should also be mentioned. What are the potential drawbacks of a magnetic nanoparticle focused AD theranostics strategy? It is important to understand and overcome this before proceeding to clinical trials. The author should discuss this in detail in the conclusions and opinions section. The reference list does not adequately cover the latest relevant representative literature. And lacks a constructive, comprehensive and critical opinion in conclusions and perspectives.
Response:
- We sincerely appreciate the valuable feedback of the reviewer, regarding the structural organization concerns about the manuscript. We agree that the subject under revision is, indeed, an extremely broad topic.
The article was constructed for general reader to first be introduced in the context of Alzheimer’s disease neuropathology and its correlation with the blood-brain barrier integrity. As a bridge for the focus main theme on magnetic nanoparticles approach for theranostics, firstly we described the current diagnosis methods for AD with the envision of magnetic resonance imaging technique. To cover that, several in vivo studies found in the literature were exhibited as evidence of magnetic NPs relevance to the topic. In a way to summarize how the MNPs features can be adapted for AD diagnosis or treatment purposes, there is a detailed description of several types of coating molecules that enable MNPs to become drug delivery systems. Due to the already referred correlation between the BBB with AD, and the fact that this barrier imposes a limitation for drugs access to the brain, we describe the application of targeted nanosystems as specific AD therapeutic platforms that exclusively transposes the BBB via RMT or AMT. This last part of the review article was complemented with a summary table.
- To extend the reader comprehension about the most recent relevant representative work on the topic, two more examples were added to the manuscript:
Line 549” Moreover, Ruan et al. developed a nanotheranostics system consisting of CUR and SPIONs encapsulated in DSPE-PEG modified with CRT (CRTIGPSVC) and QSH (QSHYRHISPAQV) peptides, which promotes specific binding to TfR and early Aβ plaques in the brain, respectively (SDP@Cur-CRT/QSH). SDP@Cur-CRT/QSH nanosys-tem enables efficient CUR delivery to the brain for sensitive AD diagnosis and amyloid plaque clearance. It demonstrates peptide-targeted BBB penetration, precise delivery to Aβ plaques for sensitive therapeutic monitoring via MRI, and cognitive improvement attributed to neuroprotection and neurogenesis induced by BDNF. Additionally, the proposed nanosystem inhibits Aβ plaques burden through the inhibition of NLR Family Pyrin Domain Containing 3 (NLRP3) inflammasomes [108].”
Line 713 “Although the mentioned examples are focused on RMT transcytosis with lig-and-specific receptors, such as LDLR, TfR, and LfR, other methods have been employed to allow BBB penetration of nanoparticle-based Alzheimer's therapeutics, via the RMT. Notably, the research conducted by Liu et al. has explored an alternative approach where they developed a dual-targeted magnetic mesoporous silica nanoparticle coated with hyaluronic acid (HA), a non-immunogenic glycosaminoglycan, that is recognized by the CD44 surface receptor. As a dual targeted Aβ clearance system, the HA coated magnetic mesoporous silica nanoparticle was further functionalized with an anti-Aβ42-targeting antibody 1F12 (HA-MMSN-1F12), to capture Aβ42 peptides. In vivo experiments reveal that the group was able to produce non-toxic NPs that accumulate in the brain and de-graded Aβ42 aggregates, consequently reducing neuroinflammation and improving memory deficits [146].”
[108] Y. Ruan et al., “Highly sensitive Curcumin‑conjugated nanotheranostic platform for detecting amyloid‑beta plaques by magnetic resonance imaging and reversing cognitive deficits of Alzheimer’s disease via NLRP3‑inhibition,” J. Nanobiotechnology, vol. 30, no. 322, pp. 1–21, 2022, doi: 10.1186/s12951-022-01524-4.
[146] N. Liu, X. Liang, C. Yang, S. Hu, Q. Luo, and H. Luo, “Dual-Targeted magnetic mesoporous silica nanoparticles reduce brain amyloid-β burden via depolymerization and intestinal metabolism,” Theranostics, vol. 12, no. 15, pp. 6646–6664, 2022, doi: 10.7150/thno.76574.
- Following the suggestion of the reviewer, a short sentence was added to the manuscript mentioning liquid biopsy techniques and blood based biomarkers with reference to other authors detailed work on the topic. [39, 40]
[39] O. Hansson et al., “The Alzheimer’s Association appropriate use recommendations for blood biomarkers in Alzheimer’s disease,” no. June, pp. 2669–2686, 2022, doi: 10.1002/alz.12756.
[40] K. Blennow and H. Zetterberg, “Biomarkers for Alzheimer’s disease: current status and prospects for the future,” no. 284, pp. 643–663, 2018, doi: 10.1111/joim.12816.
- The concerned associated with the potential drawbacks of MNPs for AD, such as toxicity, additional information was included in the conclusion section, as follow:
Line 36 on Conclusion section: “Despite of the work in here revised, researchers are continually working to address limitations associated with the in vivo use of magnetic nanoparticles. One of the main concerns is their potential for toxicity associated with the nanoparticle features, that may affect their biocompatibility, biodistribution and clearance. Addressing these challenges requires personalized nanoparticles’ synthesis and surface modifications to enhance safety and reduce size-dependent effects. Scaling up production, conducting rigorous clinical trials, and evaluating safety profiles are necessary for successful human implementation.
In summary, targeted multifunctional magnetic nanosystems hold immense promise for advancing AD theranostics. These specialized nanoparticles offer a multifaceted approach to tackling the challenges associated with the ailment, ultimately aimed at enhancing the quality of life for patients in the fight against AD.”

Reviewer 4 Report
Dear Authors,
The manuscript is really well written. The work makes a significant contribution to the field of AD theranostics, particularly by proposing MNPs as a novel neurotargeted method, and it also provides insights or viewpoints that go beyond the existing knowledge. Specific studies/experiments or applications of MNPs in AD theranostics were offered to substantiate the recommended strategies.
I only have one suggestion for the authors. It would have been better if you could have described the toxicity issues with these nanotechnology-based approaches in a separate paragraph just before the conclusion paragraph.
Author Response
Dear Krongkarn Watakulsin,
Thank you for your email and editorial decision on 27/08/2023 inviting us to submit a revised manuscript that addresses all the points raised by the reviewers.
We were pleased to see that reviewers consider that the work contributes to the field, is well organized and comprehensively described, deserving publication in Pharmaceutics. The authors would like to thank the reviewers for the helpful and highly relevant comments to improve the manuscript.
Please find below a point-by-point response to the reviewers’ comments.
REVIEWER 4
Dear Authors,
The manuscript is really well written. The work makes a significant contribution to the field of AD theranostics, particularly by proposing MNPs as a novel neurotargeted method, and it also provides insights or viewpoints that go beyond the existing knowledge. Specific studies/experiments or applications of MNPs in AD theranostics were offered to substantiate the recommended strategies.
I only have one suggestion for the authors. It would have been better if you could have described the toxicity issues with these nanotechnology-based approaches in a separate paragraph just before the conclusion paragraph.
Response: We acknowledge the revisor’s suggestion and a part of the “Conclusions” was re-written accordingly, as follows:
Line 26 on Conclusion Section: “Overall, the integration of diagnostic and therapeutic functions into a single system allows for a comprehensive approach to AD theranostics. Theranostics’ NPs offer the potential for personalized medicine, where diagnosis, targeted drug delivery, and real-time monitoring of treatment response can be achieved simultaneously. This could revolutionize the way AD is diagnosed and managed, potentially leading to more effective and tailored treatments for patients. Nevertheless, future research should focus on enhancing the selectivity of delivery systems, improving specificity in targeting ligands to minimize off-target effects, and increasing drug delivery efficiency. Additionally, exploring the use of external stimuli, such as magnetic fields, to trigger con-trolled drug release from nanosystems can further enhance their therapeutic potential.
Despite of the work in here revised, researchers are continually working to address limitations associated with the in vivo use of magnetic nanoparticles. One of the main concerns is their potential for toxicity associated with the nanoparticle features, that may affect their biocompatibility, biodistribution and clearance. Addressing these challenges requires personalized nanoparticles’ synthesis and surface modifications to enhance safety and reduce size-dependent effects. Scaling up production, conducting rigorous clinical trials, and evaluating safety profiles are necessary for successful human implementation.
In summary, targeted multifunctional magnetic nanosystems hold immense promise for advancing AD theranostics. These specialized nanoparticles offer a multifaceted approach to tackling the challenges associated with the ailment, ultimately aimed at enhancing the quality of life for patients in the fight against AD.”

Round 2
Reviewer 3 Report
I have reviewed this revised manuscript. The authors have well improved the manuscript and have properly addressed my concerns. So, this work can be recommended now to be accepted for publication in Pharmaceutics.
no